# Diabetes and Heart Failure: A Literature Review, Reflection and Outlook

**DOI:** 10.3390/biomedicines12071572

**Published:** 2024-07-15

**Authors:** Xiya Li, Xiaoyang Zhou, Ling Gao

**Affiliations:** Department of Endocrinology, Renmin Hospital, Wuhan University, Wuhan 430060, China; lxy958619764@outlook.com (X.L.); xiaoyangzh@whu.edu.cn (X.Z.)

**Keywords:** heart failure, diabetes, left ventricular ejection fraction, antidiabetic drugs, SGLT2 inhibitor, randomized controlled trial

## Abstract

Heart failure (HF) is a complex clinical syndrome caused by structural or functional dysfunction of the ventricular filling or blood supply. Diabetes mellitus (DM) is an independent predictor of mortality for HF. The increase in prevalence, co-morbidity and hospitalization rates of both DM and HF has further fueled the possibility of overlapping disease pathology between the two. For decades, antidiabetic drugs that are known to definitively increase the risk of HF are the thiazolidinediones (TZDs) and saxagliptin in the dipeptidyl peptidase-4 (DPP-4) inhibitor, and insulin, which causes sodium and water retention, and whether metformin is effective or safe for HF is not clear. Notably, sodium-glucose transporter 2 (SGLT2) inhibitors and partial glucagon-like peptide-1 receptor agonists (GLP-1 RA) all achieved positive results for HF endpoints, with SGLT2 inhibitors in particular significantly reducing the composite endpoint of cardiovascular mortality and hospitalization for heart failure (HHF). Further understanding of the mutual pathophysiological mechanisms between HF and DM may facilitate the detection of novel therapeutic targets to improve the clinical outcome. This review focuses on the association between HF and DM, emphasizing the efficacy and safety of antidiabetic drugs and HF treatment. In addition, recent therapeutic advances in HF and the important mechanisms by which SGLT2 inhibitors/mineralocorticoid receptor antagonist (MRA)/vericiguat contribute to the benefits of HF are summarized.

## 1. Background

HF is a multifaceted clinical syndrome, associated with an impaired quality of life, frequent hospitalizations and increased mortality [1]. And DM is a major risk factor for HF, which, in turn, exacerbates the metabolic disturbances and complications of DM, with a worse prognosis for patients [2]. In the Framingham Heart Study, DM resulted in a nearly twofold increase in the risk of HF in men and a fourfold increase in women [3]. In terms of possible mechanisms, DM may cause myocardial ischemia through microcirculatory disturbances and macrovascular lesions and may directly adversely affect the myocardium [4]. Hyperglycemia, hyperinsulinemia and insulin resistance can lead to fatty acid overload in cardiomyocytes and lipid deposition in the epicardium, resulting in an altered cardiac structure and impaired myocardial energy metabolism [5,6]. Advanced glycosylation products (AGEs) are deposited in endothelial cells and cross-link with myocardial collagen, worsening inflammation and oxidative stress [7]. Following a notable increase in interest in the role of antidiabetic drugs for DM in HF, a number of outcome trials for antidiabetic drugs are being demonstrated. This review will summarize the pathophysiology of DM and HF interactions, as well as antidiabetic drugs and emerging pharmacological treatments for HF.

## 2. The Pathogenesis of Heart Failure and Its Treatment

HF is primarily characterized by myocardial remodeling due to increased mechanical stress, with alterations in neural, humoral, endocrine, inflammatory, immune and energy metabolism [8]. Hyperglycemia, insulin resistance and hyperinsulinemia are the main physiological disorders of DM contributing to functional and structural disorders of the cardiovascular system through multiple pathways, finally leading to various pathological conditions, such as myocardial infarction, myocardial hypertrophy, coronary artery disease, myocardial ischemia and hypertension, and, ultimately, leading to HF [8,9] (Figure 1). Numerous underlying risk factors of HF have been identified over the decades, including hypertension, diabetes, obesity, smoking, alcohol abuse, etc. [10] However, suboptimal primary prevention of HF persists, and the incidence of HF remains high compared to ischemic heart disease and stroke [11].

## 3. Different Pathogenesis of Two Traditional Types of Heart Failure

Based on the left ventricular ejection fraction (LVEF), HF has three types: heart failure with reduced ejection fraction (HFrEF; EF < 40%), heart failure with mid-range ejection fraction (HFmrEF; EF: 40–49%), and heart failure with preserved ejection fraction (HFpEF; EF ≥ 50%). However, clinical and epidemiologic data and experimental data focus more on HFrEF and HFpEF, and this review also focuses on the two (Table 1). HFrEF is usually preceded by acute or chronic myocardial cell loss due to ischemia, genetic mutations, myocarditis or valvular disease [12]. Additionally, HFrEF is often associated with neurohormonal activation, which can further exacerbate myocardial dysfunction and adverse remodeling. In contrast, the development of HFpEF is closely related to the presence of co-morbidities [13], including obesity, diabetes, anemia, hypertension, chronic obstructive pulmonary disease (COPD), etc. A landmark invasive study showed significant impaired active diastolic function and increased passive stiffness in patients with HFpEF [14] and even concomitant adverse remodeling and dysfunction of the atrial and vascular systems [15]. Chronic low-grade systemic inflammation results from the accumulation of metabolic risk factors, simultaneously leading to inflammation in HFpEF. However, the inflammatory response in HFrEF is mostly a consequence of myocardial cell injury [16]. In addition, older women are more likely to be affected by HFpEF. However, HFrEF is more prevalent in men due to their susceptibility to myocardial infarction [17]. Notably, though medical advances have led to the development of effective and specific treatments for HFrEF by targeting the neurohumoral axis, effective drugs to treat HFpEF are currently lacking [18].

## 4. Diabetes Increases the Risk of Heart Failure

HF affects approximately 64 million people worldwide [19,20]. Sadly, nearly half of all HF patients die within five years of diagnosis [21]. The prevalence of DM in patients with HF ranges from approximately 25% to 40% (vs. 10.5% in general population) [22]. DM and hypertension are independent risk factors for HF, and the co-existence of the two significantly increases the risk of HF and cardiomyopathy [23]. The Heart and Soul study followed 839 patients with stable coronary artery disease and no signs of HF at baseline, finding that the risk of developing HF was twice as high in diabetic compared to nondiabetic subjects [24]. A retrospective cohort study showed that the incidence of HF was 30.9/1000 person-years in diabetic participants, which was higher than the incidence of 12.4/1000 person-years in nondiabetic participants [25]. According to a Medicare study, the mortality for people with type 2 diabetes mellitus (T2DM) and HF was 32.7/1000 person-years, compared to 3.7/1000-years for normoglycemic HF patients [26]. While few studies directly compare the prevalence and incidence of DM in HFrEF and HFpEF patients, in a study of inpatients with HF, the prevalence of DM was approximately 40% in both HFrEF and HFpEF patients [27].

### 4.1. Mechanism of Heart Failure Due to Diabetes Mellitus

The pathophysiological mechanisms between DM and HF are complex and multifactorial. Hyperglycemia, insulin resistance and hyperinsulinemia all seem to provoke and perpetuate the progression of HF, though the exact mechanisms are unclear [28]. Excessive production and accumulation of AGEs in plasma and vascular tissues lead to arterial stiffness and reduced elasticity [29]. Meanwhile, AGE accumulation triggers the production of reactive oxygen species (ROS), leading to myocardial and microcirculatory inflammation, mitochondrial dysfunction and myocardial apoptosis. Additionally, chronic hyperglycemia and insulin resistance may cause direct damage to the myocardium, along with the associated metabolic abnormalities, promoting the development of atherosclerosis and vascular damage. Hyperglycemia contributes to disturbed energy metabolism, inappropriate lipid deposition in extra-adipose tissues (including epicardium) [30] and lipotoxicity, provoking cardiomyocyte damage and triggering myocardial stiffness and ischemia [31]. Eventually, HF leads to renin–angiotensin–aldosterone system (RAAS) and sympathetic nervous system activation. When coupled with the typical Western diet, obesity and fatty liver, HF aggravates insulin resistance, and thus begins a vicious cycle [32] (Figure 2).

### 4.2. Diabetes and Risk Factor Management

Many studies have attempted to predict HF through markers of DM or prevent it by intervening in the course of DM. The UKPDS findings suggest that the mean value of HemoglobinA1c (HbA1c) is a predictor of cardiovascular events [33], and an early 1% reduction in HbA1c increases the long-term survival benefit by threefold. In addition, inflammatory biomarkers (TNFR-1a, osteopontin), N-terminal pro-B-type natriuretic peptide (NT-proBNP) and angiopoietin may predict cardiac remodeling, inflammation and fibrosis in patients with HF and DM [34]. A cohort study based on the Swedish National Diabetes Registry database, where risk factors (HbA1c level, VLDL level, proteinuria, smoking, hypertension) were controlled, showed that the risk of death, myocardial infarction or stroke significantly decreased, while the excess risk of HHF remained and was only slightly reduced at HbA1c less than 53 mmol/L (equivalent to 7.0%) [9]. In terms of treatment, a meta-analysis of four studies, ACCORD, ADVANCE, UKPDS33 and VADT, showed that intensive glycemic control merely leads to a small but significant cardiovascular benefit in the short to medium term but not in all-cause mortality and cardiovascular mortality [9]. The failure of many clinical trials in the past may have been due to the fact that conventional antidiabetic drugs may have additional harmful effects on the heart [35]. However, nowadays, the therapeutic role of SGLT2 inhibitors in HF across the entire spectrum of ejection fraction has shown robust evidence of a clinical benefit [36].

### 4.3. Hypoglycemia and Heart Failure

T2DM and hypoglycemia are independent risk factors for MACE and mortality in HF patients [37]. Hypoglycemia induces acute cardiovascular events, accelerates chronic complications and increases all-cause mortality [38]. In terms of the mechanism, hypoglycemia triggers counter-regulatory defense reactions, activating the renin–angiotensin system, leading to cardiac arrhythmias and hemodynamic changes [39]. It also activates inflammatory responses, platelets, macrophages and coagulation factor VIII, causing endothelial dysfunction, coagulation abnormalities and promoting atherosclerosis [40]. Additionally, it directly harms the heart by affecting myocardial energy metabolism. Notably, rebound hyperglycemia amplifies inflammation, exacerbating endothelial dysfunction and thrombus activation [41]. Major trials, like ACCORD, ADVANCE, ORIGIN and VADT, link hypoglycemia to increased cardiovascular risk and mortality [39,42,43]. The SAVOR study associates increased HF hospitalization with severe hypoglycemia from sulfonylurea combinations [44]. While the DAPA-HF study reported zero major hypoglycemic events with dapagliflozin in HF patients [45], emphasis should be placed on medications associated with an increased risk of hypoglycemia, such as insulin and sulfonylureas.

## 5. Pharmacological Treatment of Heart Failure

Prior to 2010, many randomized controlled trials (RCTs) had been conducted in the field of chronic HF, establishing the classic golden triangle treatment model of angiotensin-converting enzyme inhibitor (ACEI)/angiotensin receptor blocker (ARB), beta-blocker (BB) and MRA. After 2019, many RCTs were carried out on new drugs for HF, such as angiotensin receptor-neprilysin inhibitor (ARNI), SGLT2 inhibitors, ivabradine (Iva), etc. PARADIGM-HF suggested that ARNI represented by Sacubitril valsartan is recommended as the drug of choice for the treatment of HFrEF patients [46]. Vericiguat is a soluble guanylate cyclase (sGC) stimulator, approved for use in patients with HFrEF in several countries in 2021 and 2022, reducing the risk of HHF or the need for emergency intravenous diuretic therapy [47]. Finerenone, a third-generation mineralocorticoid receptor antagonist, shows superior renal-cardio protection. Since its first approval in the United States in 2021, it has been unanimously recommended by many domestic and international authoritative guidelines [48]. A new quadruple, ARNI/ACEI/ ARB, SGLT2 inhibitors, β-blocker and MRA, was formed in 2021, shown to provide sustained benefits [49], such as reducing the risk of cardiovascular death or HHF by 62% through early application [36]. While evolving from the golden triangle to the new quadruple therapy, the studies conducted have focused on HFrEF, even if ARNI and SGLT2 inhibitors have shown promising results in treating HFpEF, primarily by reducing HHF. Notably, patients with cardiovascular diseases with low blood pressure and fast heart rates are subjected to various limitations in the use of drugs such as ARNI and β-blockers. However, SGLT2 inhibitors have looser requirements for blood pressure and cardiac function and are better tolerated and safer. SGLT2 inhibitors are being studied in large clinical trials as other promising options [50], though we are still waiting for drugs to significantly prolong survival in HFpEF patients.

## 6. Antidiabetic Drugs and Heart Failure

DM is an independent risk factor for HF, and, ideally, effective treatment for one should benefit the other without causing harm (Table 2).

However, some antidiabetic drugs have no significant effect on HF and may even act negatively. Conventional TZDs mainly activate peroxisome proliferator-activated receptors γ(PPARγ), but overactivation often leads to significant water and sodium retention [59]. The RECORD Study and PROactive Study revealed that TZDs significantly increase the risk of HF and mortality [60]. In contrast, chiglitazar sodium, a full agonist of PPAR, has not been associated with HF in studies and has lower incidences of edema, fractures and weight gain [61]. A meta-analysis of 115 trials showed that sulfonylureas were associated with increased mortality [62]. Retrospective studies suggest insulin is associated with worse outcomes in HF patients with T2DM, but RCTs have not shown that [63]. Currently, sulfonylureas and insulin are used as second- or third-line therapies in T2DM with HF, though their safety remains controversial [64]. The MeRIA study found that acarbose reduced the absolute risk of HF in T2DM patients by up to 45% [65]. The 2013 SAVOR-TIMI 53 study showed that saxagliptin, a DPP-4 inhibitor, was cardiovascular safe compared to placebo but increased the risk of HHF [66]. However, the EXAMINE trial found that cardiovascular outcomes were similar between the alogliptin and placebo groups [67]. Meta-analyses and animal studies have shown conflicting evidence on DPP-4 inhibitors.

Some other antidiabetic drugs, however, show a more favorable outcome. The use of metformin in T2DM patients with HF has been a long-standing concern due to the potential risk of lactic acidosis [68]. However, metformin has been proven to activate AMP-activated protein kinase (AMPK), reducing cardiomyocyte apoptosis and AGE formation, enhancing mitochondrial β-oxidation of FFAs and improving myocardial function [69]. Additionally, it modulates gut microbiome changes in HF patients, particularly affecting short-chain fatty acid-producing bacteria [56]. Meta-analyses have indicated that metformin reduces the risk of all-cause and cardiovascular death, and readmission for chronic HF in T2DM patients [70,71]. Since 2007, the American Diabetes Association has recommended metformin for HF patients with normal, stable renal function [72]. Glucokinase activator (GKA), dorzagliatin, rough allosteric modulation, fully activates GK by stabilizing its active conformation, thereby improving β-cell function, reducing insulin resistance and remodeling glycemic homeostasis in patients with T2DM [73,74]. The phase III expansion study of dorzagliatin showed an improvement in time in target range (TIR) for glucose from 59.9% at baseline to 83.7% at 46 weeks of dorzagliatin treatment [75]. And while a 6% reduction in major adverse cardiovascular event (MACE) risk for every 10% increase in TIR may provide support for the cardiovascular benefits of dorzagliatin, more prospective studies are needed [76]. GLP-1 RA may improve cardiovascular outcomes by reducing appetite and body weight through central nervous system actions, decreasing hepatic steatosis to lower triglyceride and LDL-C levels [51]. In HFpEF, GLP-1RA attenuated atrial enlargement and reduced epicardial fat storage; in HFrEF, GLP-1RA attenuated adverse left ventricular remodeling and cardiac inflammation while increasing AMPK activity [77]. Recently, in the STEP-HFpEF and STEP-HFpEF DM trials, Semaglutide was superior to placebo in improving HF-related symptoms and reducing body weight in obesity-associated HFpEF [78]. SGLT2 inhibitors were effective in lowering mean HbAlc, BMI, and systolic blood pressure, as well as reducing the risk of HHF [79]. However, concerns regarding the heightened risk of genitourinary (GU) infections, particularly urinary tract infections, remain a significant barrier to their wider adoption [80].

The Grading of Recommendations Assessment, Development and Evaluation (GRADE) method of summarizing and assessing the quality of evidence is used by more than 110 organizations worldwide and has advantages over any alternative [81]. Five factors may reduce the quality of evidence: risk of bias, inconsistency, inaccuracy, indirectness and publication bias [82]. These five factors will be assessed using using the online tool, GRADEproGPT. The quality of evidence will be categorized into the following four levels: high, moderate, low and very low. Of note, the SGLT2 inhibitor is recommended in several guidelines with a high strength and quality of evidence of A, both for HFrEF and HFpEF [83,84].

## 7. SGLT2 Inhibitors and Heart Failure

Landmark trials have demonstrated the efficacy of SGLT2 inhibitors through prospective evaluations (Table 3). The EMPEROR-Reduced and DAPA-HF trials showed that SGLT2 inhibitors benefit HFrEF by reducing HHF, cardiovascular death and slowing renal function decline in chronic kidney disease (CKD) patients [85,86]. The EMPEROR-Preserved and DELIVER trials showed that SGLT2 inhibitors significantly reduced cardiovascular death and HHF in HFpEF, and modestly improved symptoms measured by the Kansas City Cardiomyopathy Questionnaire (KCCQ) [85]. Real-world database studies have also validated the clinical benefits of SGLT2 inhibitors on cardiovascular events (Table 4).

### Possible Mechanism of SGLT2i for Heart Failure

SGLT2 inhibitors lower glucose by increasing urinary glucose excretion, but their cardiovascular benefits may be independent of their hypoglycemic effects [101]. Sustained cardiovascular benefits may occur through multiple mechanisms, such as diuresis, sodium excretion, improved ventricular remodeling and cardiac function, inhibition of sodium-hydrogen exchangers, improved myocardial energy metabolism, reduced sympathetic activity, anti-inflammatory, and antioxidant activities [102] (Figure 3). The EMPA-REG OUTCOME trial showed that about 50% of the cardiovascular benefit is due to elevated hematocrit. Notably, after dapagliflozin treatment in T2DM patients, Erythropoietin (EPO) levels still increased after 2 months, suggesting that elevated hematocrit is due to both diuretic-induced hemoconcentration and increased EPO [103]. In addition, SGLT2 inhibitors protect vascular endothelial cells by enhancing NO synthesis and preventing degradation [104,105], improving ventricular remodeling and cardiac function by decreasing epicardial fat [106]. Dapagliflozin directly inhibits Sodium Hydrogen Antiporter (NHE-1), reduces intracellular Na ion concentration, and decreases glucose dependence in favor of increasing fatty acid and ketone use, an improved fuel source that requires less oxygen for cardiomyocytes. Recent studies have found SGLT2 inhibitor increased heme oxygenase expression, reduced AGEs and increased antioxidant effects of ketogenicity to alleviate oxidative stress [107,108]. SGLT 2 inhibitors can decrease activation of the transcription factor nuclear factor-κB (NF-κB), also reducing the production of pro-inflammatory cytokines and chemokines [109]. Interestingly, SGLT2 inhibitors mediated the doubling of ketone levels more significantly in diabetic patients, blocking the NLPR 3 inflammasome [110]. Additional cardiac benefits may be mediated through weight loss, reduced insulin resistance, lower serum uric acid levels, etc. [111]. In conclusion, establishing the exact mechanism of benefit is key to understanding SGLT2 inhibitors and may also provide a richer pathway to fully understand the pathophysiology of HF and the potential therapeutic approaches.

## 8. Finerenone—Non-Steroidal Mineralocorticoid Receptor Antagonists in Heart Failure

Non-aldosterone-mediated activation of the mineralocorticoid receptor (MR) promotes reactive oxygen species generation and mediates tissue inflammation and fibrosis, leading to myocardial hypertrophy, ventricular remodeling, glomerular hypertrophy and glomerulosclerosis, ultimately causing renal-cardiac adverse outcomes [112,113]. Specific MRA has become a major focus of research and development. Finerenone is the first novel non-steroidal MRA approved globally for treating T2D-related CKD [114]. It acts on multiple MR sites in the glomerulus, tubule and heart, binding specifically to Ala-773 and Ser-810 residues, leading to C-terminal ligand-dependent binding. Upon binding, helix 12 of the activation domain AF2 protrudes, forming an “inactive conformation” that fully inhibits co-regulatory factor recruitment [115,116]. Additionally, finerenone does not bind to glucocorticoid, androgen or progesterone receptors, making it more selective and providing direct anti-inflammatory and antifibrotic effects [117]. Furthermore, finerenone does not cross the blood–brain barrier, has a short half-life and allows for quicker correction of hyperkalemia, balancing efficacy and safety [118,119]. In two landmark phase III studies, FIDELIO-DKD and FIGARO-DKD, finerenone demonstrated significant renoprotective effects on top of RAAS inhibitor therapy at the maximum tolerable dose, reducing cardiovascular adverse events and proteinuria [120]. The European Society of Cardiology (ESC) gave finerenone the highest recommendation for cardiovascular disease in patients with acute and chronic HF and DM [121]. In contrast, recent subgroup analysis of the FIDELITY study showed that finerenone treatment benefit was not related to the presence or absence of SGLT2 inhibitor use [114]. However, there was an effect of better renal-cardiac benefit and lower hyperkalemia in the combination treatment group compared with the placebo group (10.3% and 2.7%, respectively) [122]. Notably, since both SGLT2 inhibitors and finerenone can cause a transient decrease in the estimated glomerular filtration rate (eGFR), sequential therapy is recommended for efficacy and safety [114]. The China Expert Consensus (2023) recommended applying finerenone with an interval of at least 2–4 weeks and to closely monitor eGFR fluctuations during treatment.

## 9. Vericiguat—A Novel Drug in HF Treatment

The 2021 ESC Heart Failure Guidelines included, for the first time, a new generation of sGC agonist vericiguat [123] (Figure 4). sGC is a key enzyme that promotes the production of cyclic guanosine monophosphate (cGMP) when nitric oxide NO binds to sGC, playing a vital role in the regulation of vascular tone, contractile function of the heart and remodeling of cardiac structure [47]. On the one hand, vericiguat improves sGC receptor sensitivity to NO by stabilizing the NO-sGC binding site, resulting in increased cGMP production in cardiomyocytes and vascular smooth muscle cells [124]. On the other hand, acting on the NO-independent binding site, it directly binds to the sGC receptor and further enhances the expression of cGMP [47]. The above dual mechanism repairs the NO-sGC-cGMP signaling pathway, improves endothelial and vascular function, and reduces ventricular remodeling and myocardial hypertrophy. The VICTORIA phase III study showed that vericiguat reduced the annual absolute risk (ARR) of the primary composite endpoint by 4.2%, with a number needed to treat (NNT) of 24, meaning one primary endpoint event is prevented per year for every 24 patients treated [125]. Continued vericiguat treatment led to significant improvements in LVEF, blood NT-proBNP levels and quality of life in patients [126]. Importantly, no adverse effects on blood pressure or renal function were observed, ensuring treatment safety [127]. Notably, for patients with worsening HFrEF, even with optimal treatment with quadruple dosing, there is still a higher risk of adverse clinical events and a higher risk of HHF [128]. Based on the results of the VICTORIA trial, vericiguat was approved for the treatment of symptomatic chronic HF with an ejection fraction <45% after a worsening event in HF due to its well-tolerated, safe and reduced rate of HHF [129].

## 10. The Reflection on Choosing the Research Endpoint in the SGLT2 Inhibitor HFpEF Test

The selection of clinical trial endpoints in ongoing studies of new drugs is fraught with challenges. Clinical trials have been a crucial tool for identifying primary endpoints since the 1960s, typically described in a non-stratified manner in the study protocol. However, when multiple primary endpoints are specified without comparing and adjusting for multiplicity, it is possible for researchers to erroneously conclude that a treatment is effective if just one of the endpoints meets a threshold of 0.05 [130]. For example, the EMPEROR-Preserved study, which used SGLT2 inhibitors as a composite endpoint for cardiovascular death or HHF, was arguably the first large HFpEF trial to meet the primary endpoint, but the improvement in this endpoint was primarily due to a reduction in HHF, with no significant differences observed in total or cardiovascular mortality, quality of life or NT-proBNP levels [131].

## 11. Future Directions in Heart Failure Research

RCTs include patients with HFrEF to improve statistical efficacy, ignoring the diverse and heterogeneous phenotypes of HF patients [132]. Future studies should focus on the characteristics of the HF spectrum, moving from analyses of large, heterogeneous groups with unspecified LVEF to more individualized [133], mechanistic studies with smaller, homogeneous groups. Additionally, benefits have been shown in patients with DM and/or HF, suggesting that evaluating their potential synergy with SGLT2 inhibitors is promising [51]. Recent cohort studies indicate that co-administering GLP-1 RA and SGLT2 inhibitors reduces the risk of MACE and serious renal events compared to using either drug alone [134]. The combination of an SGLT2 inhibitor and MRA will be evaluated in patients with cardiorenal disease in the MIRACLE and CONFIDENCE trials [135]. Moreover, SGLT2 inhibitors will be applied to address causative factors leading to HF and other cardiovascular diseases. Since atrial fibrillation is significantly associated with HFpEF, SGLT2 inhibitors have been shown to reduce atrial fibrillation events in a recent analysis of adverse event reports from RCTs [136]. Additionally, starting SGLT2 inhibitors in patients with T2DM and a history of percutaneous coronary intervention (PCI) was significantly associated with reduced risks of cardiorenal outcomes and mortality, regardless of the time since the last PCI [137]. Finally, ongoing trials (DAPA-MI and EMPACT-MI) are assessing the role of SGLT2 inhibitors after myocardial infarction [84].

## 12. Conclusions

HF and DM are mutual risk factors for each other. Long-term hyperglycemia can lead to autonomic neuropathy, microcirculatory dysfunction and metabolic and energy changes. Therefore, hypoglycemic therapy has been widely researched as an important potential target for HF. Encouragingly, many antidiabetic drugs applied to HF have achieved good efficacy, such as SGLT2 inhibitor and some GLP-1 RA. HFpEF accounts for nearly 50% of cases, with hospitalization and mortality rates comparable to HFrEF, and still lacks ideal treatment, while SGLT2 inhibitors have shown significant benefit in preventing hospitalization. The potential mechanisms of the SGLT2 inhibitor cannot be attributed solely to improvements in traditional risk factors (e.g., blood pressure, glucose, volume status and renal function), and an increasing amount of experimental evidence suggests an improvement in energy efficiency and a reduction in inflammation and oxidative stress and fibrosis, which can help to reduce adverse remodeling and diastolic dysfunction. These studies will provide important insights into the key pathways involved in the cardioprotective effects associated with SGLT2 inhibitor therapy. In addition, novel drugs such as vericiguat and finerenone have been documented in several clinical trials for very promising cardiovascular benefits in HF. Hopefully, as clinical evidence increases, they will become an additional powerful tool in the treatment of patients with HF. While this review comprehensively synthesizes the current literature on DM, HF and antidiabetic drugs for HF treatment, several limitations should be noted. The literature search may have been restricted by the timeframe, language biases and specific database reliance, potentially excluding relevant high-quality studies. Additionally, this review primarily aligns with authoritative guidelines, rather than autonomously applying the GRADE methodology to assess antidiabetic drug recommendations based on evidence quality and clinical context, and to indicate the recommendation strength clearly. Moreover, long-term safety data for some drugs remain unclear due to limited clinical trial follow-up. Future research should address these limitations.

## Figures and Tables

**Figure 1 biomedicines-12-01572-f001:**
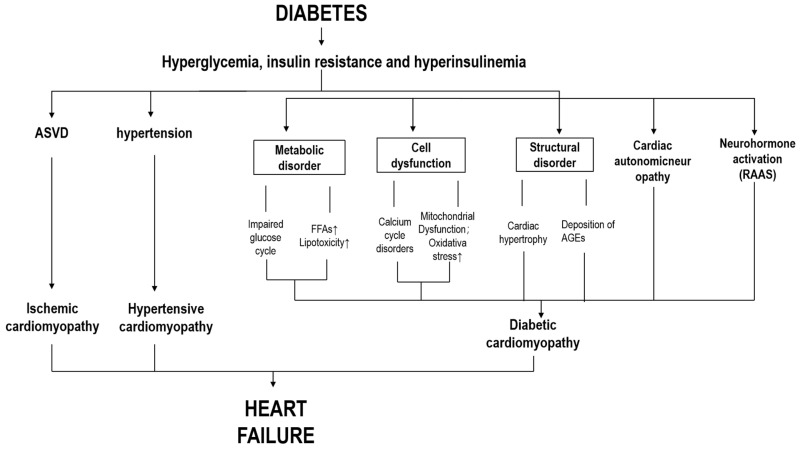
Pathophysiological mechanism of diabetic heart failure. Diabetic patients with hyperglycemia, insulin resistance, and hyperinsulinemia trigger vascular lesions, metabolic abnormalities, cellular dysfunction, and abnormal activation of the endocrine and nervous systems of the body, leading to ischemic heart disease, hypertensive heart disease, diabetic cardiomyopathy, and ultimately heart failure. ASVD, atherosclerotic vascular disease; FFAs, free fatty acids; AGEs, advanced glycosylation products; RAAS, renin–angiotensin–aldosterone system.

**Figure 2 biomedicines-12-01572-f002:**
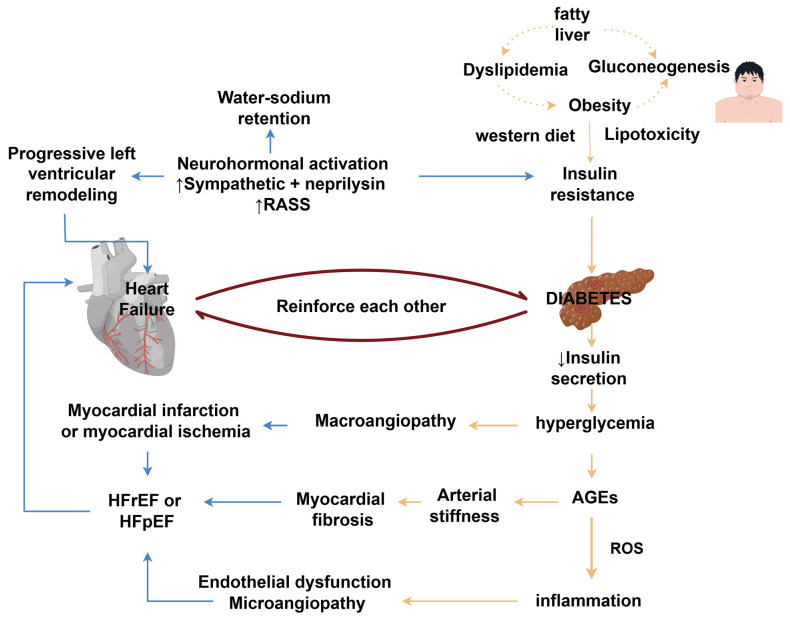
The pathological mechanism of interaction between heart failure and diabetes. Diabetes mellitus leads to hyperglycemia and insulin resistance, and the chronic hyperglycemic leads to chronic inflammation, endothelial and cardiomyocyte damage, promoting cardiac fibrosis, myocardial ischemia, cardiac remodeling, and further worsening heart failure. Heart failure leads to decreased cardiac output, sodium and water retention, activation of sympathetic nerves and the RASS system, which further reduces insulin secretion and exacerbates hyperglycemia, creating a vicious cycle. By Figdraw. RAAS, renin–angiotensin–aldosterone system; AGEs, advanced glycosylation products; ROS, reactive oxygen species; HFrEF, heart failure with reduced ejection fraction; HFpEF, heart failure with preserved ejection fraction.

**Figure 3 biomedicines-12-01572-f003:**
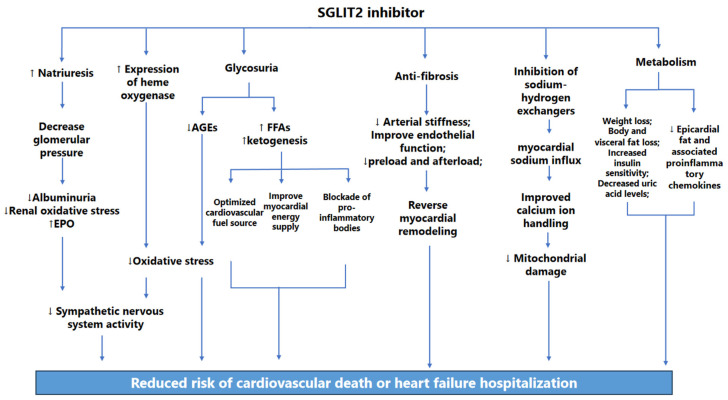
Potential mechanisms of clinical benefit of SGLT2 inhibitors. SGLT2 inhibitors exert beneficial effects on the cardiovascular system through a variety of mechanisms, including hypoglycemia, antihypertension, diuresis, weight loss, improvement of myocardial energy metabolism, reduction of oxidative stress and inflammation, protection of renal function, body weight reduction, and metabolic improvement. Recently, SGLT2 inhibitors have been found to have novel mechanisms of increased heme oxygenase expression, increased ketone bodies as a better fuel for the myocardium, and decreased epicardial fat, which together reduced risk of cardiovascular death or heart failure hospitalization. SGLT2, sodium-glucose transporter 2; EPO, erythropoietin; AGEs, advanced glycosylation products; FFAs, free fatty acids.

**Figure 4 biomedicines-12-01572-f004:**
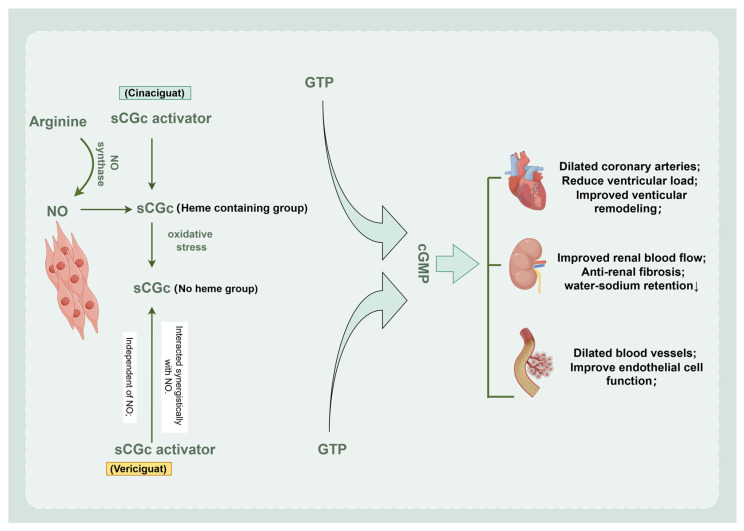
Mechanism of cardiorenal benefits of vericiguat. Vericiguat is the first sGC agonist that can directly stimulate sGC or synergize with endogenous NO to jointly increase intracellular levels of cGMP, leading to smooth muscle relaxation and vasodilation. By repairing damage to NO-sGC-cGMP, a cellular signaling pathway, the three major target organs, heart, blood vessels, and kidneys, are benefited. By Figdraw. SCG, soluble guanylate cyclase; NO, nitric oxide; cGMP, cyclic guanosine monophosphate; GTP, guanosine triphosphate.

**Table 1 biomedicines-12-01572-t001:** Difference in pathogenesis of HFpEF and HFrEF.

	HFpEF	HFrEF
Diagnostic criteria	① Symptoms/signs of HF; ② LVEF ≥ 50%; ③ Natriuretic peptide + (left ventricular hypertrophy, left atrial enlargement, cardiac diastolic dysfunction).	① Symptoms/signs of heart failure; ② LVEF < 40%.
Pathogenesis	Mostly associated with ischemic diseases, endothelial dysfunction at early stages and microvascular events more prevalent.	Altered left ventricular structure and function;Neurohormonal activation and endothelial dysfunction at all stages;
Co-morbidities	Obesity, diabetes, anemia, hypertension, chronic obstructive pulmonary disease, autoimmune disease and renal insufficiency.	Acute or chronic Cardiomyocyte loss due to prior ischemia, genetic mutation, myocarditis, or valvular disease;
Age and sex	Older women with obesity.	Men with coronary heart disease, valvular disease or uncontrolled hypertension.
Inflammation	Low-grade systemic inflammation.	Consequence of myocardial cell injury.
Treatment	Effective treatment is still lacking.	ACEIs, β-blockers, ARBs, MRAs, ARNI, If-channel blockers are recommended.

HF, heart failure; LVEF, left ventricular ejection fraction; HFrEF, heart failure with reduced ejection fraction; HFpEF, heart failure with preserved ejection fraction; ACEIs, Angiotensin-converting enzyme inhibitors; β-blockers, β-adrenergic blockers; ARBs, angiotensin receptor blockers; MRAs, mineralocorticoid receptor antagonists; ARNI, angiotensin receptor neprilysin inhibitors; If-channel, IF channel inhibitors.

**Table 2 biomedicines-12-01572-t002:** Summary of antidiabetic drug classes on heart failure [51,52,53,54,55,56,57,58].

	Class	Common Medications	Effects in Heart Failure	Clinical Practice Recommendations
Novel antidiabetic drugs	SGLT2 inhibitors	Dapagliflozin Empagliflozin	Beneficial	Symptomatic HFrEF patients are recommended regardless of DM.
	GLP-1 RA	Semaglutide	Beneficial	If SGLT2 inhibitors are contraindicated or not tolerated, consider GLP-1 RA. Recommended for HFpEF patients (BMI ≥ 30 kg/m^2^, NYHA class II-IV, LVEF ≥ 45%).
	DPP-4i	Saxagliptin	Saxagliptin harmful Others potentially harmful	Saxagliptin is not recommended for T2DM with HF.
	GKA	Dorzagliatin	Potential cardiovascular benefits	Solid protection for CHD and HF, but more RCTs are needed.
Traditional antidiabetic drugs	Metformin	Metformin	Beneficial	Recommended for stable T2DM patients with normal renal function and chronic HF; not recommended for acute or decompensated HF patients.
	Glycosidase inhibitor	Acarbose	Beneficial	Second- or third-line medication. If digoxin is used to treat HF, avoid or discontinue acarbose.
	Glinides	Repaglinide	Neutral	Second- or third-line medication. Does not increase cardiovascular events in patients with T2DM.
	Sulfonylureas	Glimepiride	Potentially harmful	Sulfonylureas are not recommended for T2DM with HF.
	Insulin	Insulin	Potentially harmful	Rapid-acting and short-acting insulin analogs appears safer, with close monitoring for potential worsening of HF.
	TZDs	Pioglitazone	Harmful	TZDs are not recommended for T2DM with HF.

SGLT2i, sodium-glucose transporter 2 inhibitor; GLP-1 RA, glucagon-like peptide-1 receptor agonists; DPP-4i, dipeptidyl peptidase-4 inhibitor; GKA, glucokinase activator; TZDs, thiazolidinedions; HFrEF, heart failure with reduced ejection fraction; HFpEF, heart failure with preserved ejection fraction; DM, diabetes mellitus; HF, heart failure; CHD, coronary heart disease; NYHA, New York Heart Association; RCTs, randomized controlled studies.

**Table 3 biomedicines-12-01572-t003:** RCT: SGLT2i on HF.

Name	Method	Result	Conclusion
The DAPA-HF study [45]	N = 4744 Dapagliflozin 10 mg vs. placebo Follow-up of 18.2 years	Dapagliflozin can reduce the risk of cardiovascular death or HHF in HFrEF patients by 26% and the risk of all-cause mortality by 17%.	Dapagliflozin reduces the risk of ventricular arrhythmia, cardiac arrest, or sudden death in patients with HFrEF.
The EMPEROR-Reduced study [86]	N = 3730 Empagliflozin 10 mg vs. placebo Follow-up of 1.5 years	Empagliflozin reduced the primary outcome (first cardiovascular death or HF with or without CKD), as well as the secondary outcome (patients’ first and recurrent hospitalizations for HF) and may slow the rate of decline in eGFR.	The use of empagliflozin in HFrEF patients is recommeded.
The DELIVER study [87]	N = 6263 Dapagliflozin 10 mg vs. placebo Follow-up of 2.3 years	Dapagliflozin significantly reduced the rate of cardiovascular deaths and HHF by 18%, regardless of left ventricular function.	Dapagliflozin reduce the risk of the primary composite endpoint.
The PRESERVED-HF study [88]	N = 324 Dapagliflozin 10 mg vs. placebo Follow-up of 12 weeks	Dapagliflozin significant benefit in symptoms, physical limitations and 6-min walk test with 12 weeks of dapagliflozin.	Within 12 weeks, dapagliflozin significantly improved symptoms and physical activity limitations in patients with HFpEF.
The CAMEO-DAPA study [89]	N = 38 NYHAII/III, LVEF > 50% and PCWP patients were randomly divided into the dapagliflozin group and placebo group. Follow-up of 24 weeks	Resting PCWP decreased by 3.5 mmHg (*p* = 0.029); Exercise PCWP decreased by 6.1 mmHg (*p* = 0.019)	Dapagliflozin offers potential benefits for HFpEF.
The EMPEROR-Preserved study [90]	N = 5988 Empagliflozin 10 mg vs. placebo, Follow-up of 26.2 months	Empagliflozin significantly improved cardiovascular death and HHF in patients with HFpEF, the treatment effect of empagliflozin was not affected by heart rate.	Empagliflozin significantly reduced the rate of cardiovascular death or HHF composite endpoint events compared to placebo, and independent of the presence of comorbid diabetes.
The EMPA-RESPONSE-AHF study [91]	N = 80 Empagliflozin 10 mg vs. placebo Follow-up of 60 days	Empagliflozin significantly reduced the composite endpoint events including worsening heart failure during hospitalization, HF rehospitalization, and death.	Empagliflozin is safe and well-tolerated in the treatment of acute decompensated HF.
the EMPA-REG OUTCOME study [92]	N = 7020 Empagliflozin 10 mg vs. Empagliflozin 25 mg vs. placebo	Empagliflozin significantly reduced the risk of 3P-MACE in patients by up to 14% and the risk of cardiovascular death by up to 38% on top of standard therapy.	Treatment with empagliflozin (10 mg) significantly prolonged survival in patients of all ages.
The EMBRACE-HF study [91]	N = 65 Empagliflozin 10mg vs. placebo Follow-up of 12 weeks	The empagliflozin group significantly reduced mean pulmonary artery diastolic pressure at 8–12 weeks.	Empagliflozin significantly reduced pulmonary artery diastolic blood pressure.
The EMPULSE study [93]	N = 530 Empagliflozin 10 mg vs. placebo Follow-up of 90 days	Patients treated with empagliflozin were 36% more likely to achieve clinical benefit, and no treatment heterogeneity.	Patients hospitalized with acute HF were significantly more likely to have clinical benefit within 90 days with empagliflozin.
The EMPAG-HF study [94]	Empagliflozin 25 mg vs. placebo Follow-up on days 6–10 (at discharge) and after 30 days	Treatment of acute decompensated HF with empagliflozin does not result in additional renal injury.	Empagliflozin was safe and well tolerated.
The VERTIS CV study [95]	N = 8246 Ertugliflozin 5 mg vs. Ertugliflozin 15 mg vs. placebo Follow-up of 3.5 years	The risk of 3P-MACE was noninferior in the Ertugliflozin (5 and 10 mg) groups compared with the placebo group.	Ertugliflozin reduces the risk of first HF and all HF after enrollment, which is not influenced by the presence or absence of a history of HF at baseline or LVEF.
The CHIENT-HF study [96]	N = 476 HF Canagliflozin 100 mg vs. placebo	The benefit of canagliflozin in significantly improving patient symptom regardless of EF or T2DM.	Canagliflozin not only improves prognosis, but also significantly improves symptoms, function and quality of life.

HF, heart failure; HHF, hospitalization for heart failure; EF, ejection fraction; LVEF, left ventricular ejection fraction; NYHA, New York Heart Association; PCWP, pulmonary capillary wedge pressure; HFrEF, heart failure with reduced ejection fraction; HFpEF, heart failure with preserved ejection fraction; CKD, chronic kidney disease; T2DM, type 2 diabetes mellitus.

**Table 4 biomedicines-12-01572-t004:** Real world: SGLT2 inhibitor on HF [97,98,99,100].

Name	Populations Included	Method	Conclusion
CVD-REAL study	Patients treated with any SGLT-2i versus other antidiabetic drugs in 6 countries (USA, Norway, Denmark, Sweden, Germany and UK).	Using medical claims, hospital records and national registry data, HHF and mortality were estimated by country and aggregated to determine a weighted effect size.	Treatment with SGLT-2 inhibitor was associated with lower incidence of HHF and mortality, with no heterogeneity in outcomes between countries.
Scandinavian register based on cohort study	National registry data from Denmark, Norway, and Sweden including 19% of the population with a history of major cardiovascular disease and 6% with a history of HF.	83% were on dapagliflozin, 16% empagliflozin, and 1% canagliflozin, compared SGLT-2 inhibitor versus DPP-4is on heart failures hospitalization.	SGLT-2i reduced the risk of primary heart failure by 34%.
Real-world study in Taiwan	12,681 patients with T2DM who were new to SGLT2 inhibitors were included, of which 5812 were on dapagliflozin and 6869 were on empagliflozin.	Through analysis of multi-institutional electronic medical records (Chang Gung Research Database, Taiwan), the risk of cardiovascular events was compared and analyzed between the two groups.	In terms of risk of cardiovascular events, both risks were similar, but dapagliflozin was significantly better than empagliflozin in reducing HF.

HF, heart failure; HHF, hospitalization for HF; SGLT2, sodium-glucose transporter 2; T2DM, type 2 diabetes mellitus.

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
