# Peer review of "Diabetes and Heart Failure: A Literature Review, Reflection and Outlook"

_biomedicines, 2024, doi:10.3390/biomedicines12071572_

Round 1

Reviewer 1 Report

Comments and Suggestions for Authors

The article is very interesting and the topic addressed is actual and of interest to clinical practice. I have some suggestions before publication, to improve the quality of the manuscript.

1. The opening sentence attempts to define HF but could benefit from a clearer explanation of how DM impacts HF specifically.

2. The article lists several hypoglycemic drugs, but the discussion is somewhat vague.

3. The section jumps between drug effects and broader disease interactions without a clear logical flow. Grouping information by drug type or by the effect (positive vs. negative) could help.

4. The article claims effects for various drugs on HF outcomes but does not provide supporting data or literature references.

5. Some sentences are lengthy and contain multiple ideas, which could be split for better readability and impact.

By addressing these points, the manuscript could present a more compelling, well-supported, and reader-friendly narrative, making it a valuable contribution to the literature on the intersection of HF and DM.

Comments on the Quality of English Language

Minor editing of English language required. 

Author Response

Dear reviewer:

Thank you very much for your consideration and for the review’s comments on our manuscript entitled “Diabetes and Heart Failure: Review, Reflection and Outlook”. we are very grateful receive your comments which are significantly helpful for improving our paper. As you are concerned, there are several problems that need to be addressed. According to your nice suggestions, we have made corrections to our previous draft, the detailed corrections are listed below: Reviewer comments are given below in black font, with specific questions numbered. Our responses are given in blue font, and changes/additions to the manuscript are given in YELLOW highlighted text.

I hope that revised manuscript is suitable for publication in Biomedicines. Thank you again for your help with the manuscript. I am more than happy yo provide any further information you may need.

Sincerely yours,

  1. The opening sentence attempts to define HF but could benefit from a clearer explanation of how DM impacts HF specific

ANSWER:  Thanks for the suggestion, we sincerely appreciate the valuable comments. We have re-written this part, including the exact location where the change can be founded in the opening sentence in the revise manuscript.

  1. The article lists several hypoglycemic drugs, but the discussion is somewhat vague.

ANSWER:  Thank you for your comment. We tried our best to articulate more clearly the effects of hypoglycemic drugs in the treatment of heart failure. Here we do not list the large one-paragraph changes, but the main idea and framework are marked in yellow on page 7 of the revised paper.

  1. The section jumps between drug effects and broader disease interactions without a clear logical flow. Grouping information by drug type or by the effect (positive vs. negative) could help.

ANSWER:  We think this is an excellent suggestion. We have reorganized paragraphs in this part according to the Reviewer's suggestion.

  1. The article claims effects for various drugs on HF outcomes but does not provide supporting data or literature references.

ANSWER: We sincerely appreciate the valuable comments. We have checked the literature carefully and added more references on hypoglycemic drugs on HF outcomes, which you can see on page 7 and in Table 2 of the revised manuscript.

  1. Some sentences are lengthy and contain multiple ideas, which could be split for better readability and impact.

ANSWER: Thanks for your suggestion, we have tried our best to make our article sentences more fluent and understandable in the revised manuscript. We appreciate for reviewers’ warm work earnestly and hope that the correction will meet with approval.

Thanks to the professional comments again that point out the above problems. The authors hope these explanations would answer your doubts.

Reviewer 2 Report

Comments and Suggestions for Authors

This is a very nice and timely review article about the association between heart failure (HF) and type 2 diabetes (T2D), with a special focus on the efficacy and safety of hypoglycemic drugs and their role in the modern therapy of HF. Indeed, these two dieases are closely linked and some T2D are also beneficial for HF even independently of their glucose lowering effects or antidiabetic action in general. This type of review paper is highly apreciated. I could detect couple of issues to be resorved before publication.

1.     The text and tables/figure contain some abbreviation which are not properly explained in the text. Please check and fix carefully. Some examples are- ASCVD and RAAS (Figure 1), CHD (page 4), HHF (pages 8, 12), MACE (page 8) et c.

2.     There is a mistake in Table 1 in the tretment section – information for HFpEF and HFrEF is essentially swaped here.

3.     Table 2 is duplicated. I also do not completely understand why the authors indicate here that the effect of Semaglutide in HF is «neutral». Later on page 8, they correctly describe the results of a recent STEP-HFpEF trial showing that this drug is beneficial in HFpEF.

4.     Clinical benefit and indications for vericiguat have not been accurately described. They talk about benefit for HF in general, not mentioning that this drug is especially useful in patients with «worsening HF».

5.     There are some typos in FIgure 3, e.g. «netablism», «acidlevels» 

Comments on the Quality of English Language

overall very good, only minor issues detected

Author Response

Dear reviewer:

Thank you very much for your consideration and for the review’s comments on our manuscript entitled “Diabetes and Heart Failure: Review, Reflection and Outlook”. we are very grateful receive your comments which are significantly helpful for improving our paper.As you are concerned, there are several problems that need to be addressed. According to your nice suggestions, we have made corrections to our previous draft, the detailed corrections are listed below: Reviewer comments are given below in black font, with specific questions numbered. Our responses are given in blue font , and changes/additions to the manuscript are given in RED highlighted text.

I hope that revised manuscript is suitable for publication in Biomedicines. Than you again for your help with the manuscript. I am more than happy yo provide any further information you may need.

Sincerely yours,

  1. The text and tables/figure contain some abbreviation which are not properly explained in the text. Please check and fix carefully. Some examples are- ASCVD and RAAS (Figure 1), CHD (page 4), HHF (pages 8, 12), MACE (page 8) et c.

ANSWER: We sincerely thank the reviewer for careful reading. As suggested by the reviewer, we have double-checked that the text and tables/graphs contain some abbreviations and corrected them.

  1. There is a mistake in Table 1 in the treatment section – information for HFpEF and HFrEF is essentially swaped here.

ANSWER: We were really sorry for our careless mistakes.Thank you for your remind, we have corrected the mistake.

  1. Table 2 is duplicated. I also do not completely understand why the authors indicate here that the effect of Semaglutide in HF is «neutral». Later on page 8, they correctly describe the results of a recent STEP-HFpEF trial showing that this drug is beneficial in HFpEF.

ANSWER: Thanks for your careful checks. We are sorry for our carelessness. We have corrected this after a careful review of the literature.

  1. Clinical benefit and indications for vericiguat have not been accurately described. They talk about benefit for HF in general, not mentioning that this drug is especially useful in patients with «worsening HF».

ANSWER: We sincerely appreciate the valuable comments.We have added on this section in the revised manuscript after reading the relevant literature, and your valuable comments have made me handle my manuscript more seriously.

  1. There are some typos in FIgure 3, e.g. «netablism», «acidlevels»

ANSWER: We feel sorry for our carelessness in our manuscript, the typo is revised. Thanks for your correction.

Thanks to the professional comments again that point out the above problems. The authors hope these explanations would answer your doubts.

Reviewer 3 Report

Comments and Suggestions for Authors

The article is not original but this is a review.

The main concern is table 2 : the title and concept are wrong ! GLP1 Ra and sGLT2i and iDPP4are NOT hypoglycemic drugs and should be shown in a different table.

Comments on the Quality of English Language

No particular concern, to me english is OK

Author Response

Dear reviewer:

Thank you very much for your consideration and for the review’s comments on our manuscript entitled “Diabetes and Heart Failure: Review, Reflection and Outlook”. we are very grateful receive your comments which are significantly helpful for improving our paper.As you are concerned, there are several problems that need to be addressed. According to your nice suggestions, we have made corrections to our previous draft, the detailed corrections are listed below: Reviewer comments are given below in black font, with specific questions numbered. Our responses are given in blue font, and changes/additions to the manuscript are given in GREEN highlighted text.

I hope that revised manuscript is suitable for publication in Biomedicines. Thank you again for your help with the manuscript. I am more than happy yo provide any further information you may need.

Sincerely yours,

  1. The article is not original but this is a review.
    ANSWER: We would like to thank you for your professional review work, constructive comments, and valuable suggestions on our manuscript. We agree with you, here is one review which we have compiled the latest literature and summarized. And we still hope that in the future we will create our original ARTICLES to meet you and get your valuable and professional suggestions again.

2.The main concern is table 2 : the title and concept are wrong ! GLP1 Ra and sGLT2i and iDPP4 are NOT hypoglycemic drugs and should be shown in a different table.
ANSWER:We thank the reviewer for pointing out this potential confound.It is true that GLP-1 RAs, SGLT2i and DPP4i, and GKA do not belong to the traditional hypoglycemic drugs; they are a relatively new class of drugs that are not simply a class of drugs that just lower blood glucose. However, both GLP-1 RAs, which mimic the GLP-1 hormone in the body and enhance insulin secretion, and SGLT2i, which inhibit the SGLT2 protein in the kidneys and increase urinary glucose excretion, have the ultimate effect of lowering glucose. Therefore, in our review, we define them as "novel hypoglycemic drugs" and and adjusted Table 2. We hope you will accept our modification. However, if you have any other suggestions or ideas about this, we will positively make corrections and thank you again for your professional advice.

Thanks to the professional comments again that point out the above problems. The authors hope these explanations would answer your doubts.

Round 2

Reviewer 3 Report

Comments and Suggestions for Authors

Table 2 is not reviewed (Hypolycemic is WRONG)

Furthermore a GRADE method should be provided

Discuss the role oh hypos in HF 

Author Response

Dear reviewer,

Thank you for your precious time and kind comments on this review and also thank for providing this valuable opportunity to revise our manuscript. We extremely cherish this opportunity to revise and have made a lot of revisions compared with last version of manuscript followed the suggestion and comments of you. Response to your concerns and questions, our responses are given in blue font, the changes made in the revised manuscript are highlighted in GREY as detailed below:

  1. Table 2 is not reviewed (Hypolycemic is WRONG)

ANSWER: we apologize for our original writing which may mislead the reviewer. To avoid confusion, we have modified the term "hypoglycemic" to "antidiabetic" and highlighted it in DARK GREEN, and we really appreciate your rigorous consideration to our review.

  1. Furthermore a GRADE method should be provided

ANSWER: Thanks for your professional suggestion. The GRADE grading system improves the quality and credibility of evidence assessment and clinical recommendations through its transparency, standardization, clinical relevance, multidimensional assessment, flexibility, and clarity of the grading system.Your suggestions have improved the ability of our review to provide high quality evidence. This REVIEW is a literature review, we prefer to focus on overview and summarize existing studies. So we have summarized the valid recommendations for the above mentioned antidiabetic drugs in the clinical application practice guidelines in Table 2 after reading more authoritative guidelines.We briefly describe the GRADE approach in the revised REVIEW, as well as refer to multiple guidelines for grading the evidence for SGLT2 inhibitors, you can see it on page 8, line 264. Unfortunately, however, we did not have sufficient time and capacity to perform our own GRADE methodology process in this article. so we have briefly described it as starved of LIMITATIONS in the DISCUSSION section.And we promise that we will take it into consideration seriously in our future work.

  1. Discuss the role oh hypos in HF

We sincerely thank you for your valuable feed back that we have used to improve the quality of our manuscript. Since we're not sure if you accidentally made a typo due to your busy schedule, we read it as "Discuss the role of hypos in HF". We have added this part according to the Reviewer's suggestion, which you can see in Section 4.3 beginning at line 165 on page 6.

Thank you once again for your interest in our article and for providing such professional advice. From your suggestions, we can infer that you are a very meticulous and serious scholar, and your feedback has significantly impacted our revision work. We hope these explanations would answer your doubts.

Sincerely,

Corresponding author.

Round 3

Reviewer 3 Report

Comments and Suggestions for Authors

Thank you no more suggestions